# Oral Plus Topical Administration of Enrofloxacin-Hydrochloride-Dihydrate for the Treatment of Unresponsive Canine Pyoderma. A Clinical Trial

**DOI:** 10.3390/ani10060943

**Published:** 2020-05-29

**Authors:** Lilia Gutierrez, Graciela Tapia, Luis Ocampo, Minerva Monroy-Barreto, Hector Sumano

**Affiliations:** 1Departamento de Fisiología y Farmacologia, Facultad de Medicina Veterinaria y Zootecnia, Universidad Nacional Autónoma de México, Avenida Universidad 3000, Coyoacan, Mexico City 04510, Mexico; liliago@unam.mx (L.G.); farmac@unam.mx (L.O.); 2Departamento de Genética y Bioestadística, Facultad de Medicina Veterinaria y Zootecnia, Universidad Nacional Autónoma de México, Avenida Universidad 3000, Coyoacan, Mexico City 04510, Mexico; tapiadoctora@gmail.com; 3Departamento de Química Analítica, Facultad de Química, Universidad Nacional Autónoma de México, Avenida Universidad 3000, Coyoacan, Mexico City 04510, Mexico; monroy17@unam.mx

**Keywords:** unresponsive canine deep-pyoderma, enrofloxacin hydrochloride-dihydrate, treatment, clinical trial

## Abstract

**Simple Summary:**

This is the first report on the clinical use of a new derivative of enrofloxacin (enrofloxacin HCl-2H2O or enro-C) for the treatment of canine unresponsive deep-pyoderma (UDCP), utilizing a dual scheme, i.e., 10 mg/kg/day in capsules, plus the topical administration of enro-C, prepared as an 0.5% alginate gel, thrice per day. Fifty-five cases of UDCP were treated successfully in a one-year study. Mean days of treatment were 8 to 12, for either severe or very severe cases, respectively. Complete success was recorded and no recurrences after a two-month clinical follow up were registered. It is here proposed that the dual treatment, the potency of enro-C and the theoretically high concentrations of the active principle in the lesions may explain these findings. Further research is needed to define the bacteriological status of the pathogens found, and the impact of this treatment in bacterial resistance.

**Abstract:**

An outpatient clinical trial on unresponsive deep-bacterial canine pyoderma (UDCP), without a control group, is presented. The chosen treatment was implemented with a new crystal-solvate of enrofloxacin (enrofloxacin HCl-2H_2_O or enro-C), in a dual scheme, i.e., 10 mg/kg/day PO, plus its topical administration, prepared as 0.5% in an alginate gel, thrice per day. Fifty-five cases that were unsuccessfully treated previously with another antibacterial drug, were selected and then classified as severe or very severe, according to a clinical score tailored for this trial. Aerobic bacteriological cultures of skin lesions and antibacterial sensitivity tests, were performed. Hematological status, liver, and kidney functions were determined before and after treatment. A complete success was obtained in 32 severe and 23 very severe, cases. The main bacterial isolates were: *Staphylococcus intermedius* (19/99), *Staphylococcus pseudintermedius* (16/99), *Staphylococcus epidermidis* (15/99), *Staphylococcus pyogenes* (14/99), *Staphylococcus saprophyticus*, *Streptococcus* sp., and others including *Pseudomonas aeruginosa* (6/99). The average duration of treatment was 8.03 days ± 2.1 SD and 12.0 ± 2.4 days, for dogs with severe or very severe UDCP, respectively. The adverse effects caused by enro-C were inconsequential and the hematological tests showed no deviations from normality. The use of enro-C administered dually to treat UDCP, is considered safe and highly effective.

## 1. Introduction

Bacterial skin infections in dogs are among the most common conditions requiring veterinary attention [1,2,3]. An initial skin lesion might end up as superficial pyoderma or folliculitis or deep pyoderma or furunculosis/cellulitis, caused mainly by a bacterial agent. Pyoderma and accompanying inflammatory changes cause severe pruritus and facilitate self-injury and mutilation. It is important to treat the primary cause of pyoderma to avoid recurrences, such as bacterial overgrowth syndrome, juvenile cellulitis, calciphylaxis due to end-stage renal disease and hyperparathyroidism, immunomodulatory-responsive lymphocytic-plasmacytic pododermatitis, pemphigus foliaceous, pyoderma gangrenosum, and other pathologies [2]. In daily clinical practice, the suspicion of bacterial infection is often treated empirically with antibiotics, and treatment is usually successful. However, recurrences occur mainly due to premature withdrawal of the antibacterial drug based on clinical cure criterium rather than a bacteriological one. Treatment of recurrences should be based on bacterial culture and susceptibility test results. Resistance to several classes of commonly used antimicrobial drugs has been reported, for example, β-lactam antibacterial drugs [4], clindamycin [5,6], and fluoroquinolones [4,7]. It has been noted that bacterial resistance to chemotherapy is increasing [2], as evidenced by the increased resistance of *Staphylococcus intermedius*, *Staphylococcus pseudointermedius*, *Staphylococcus aureus* [8,9], and *Staphylococcus schleiferi,* indicates [10]. Additionally, methicillin-resistant *Staphylococcus aureus* (MRSA) identical to human EMRSA-15 have been found in dogs and hospital staff of veterinary clinics in some parts of the world [11,12]. Antibacterial derivatives of cephalosporin are usually prescribed first to treat pyoderma. If unsuccessful, fluoroquinolones are used as rescue antibacterial drugs and it has been shown that pradofloxacin [13], ibafloxacin, marbofloxacin [14], and enrofloxacin [15] can be effective options. Optimal use of fluoroquinoles require appropriate dosage and good quality fluoroquinolones considering that the maximum serum concentration (C_MAX_) after administration is a key feature for clinical success, i.e., for enrofloxacin—a concentration-dependent antibacterial drug—a C_MAX_/MIC ≥ 10–12 ratio must be met [16,17]. To treat canine pyoderma, this ratio seems to be obtainable with high doses of enrofloxacin administered orally (i.e., 10 mg/kg) [18]. However, in some cases, the different qualities of tablet manufacturing and the active ingredients used [19] can alter oral bioavailability, preventing the achievement of the above-mentioned ratio [20]. Reduced bioavailability of enrofloxacin can also be caused when administered with certain foods [21], or due to the lack of bioequivalence of pharmaceutical preparations, as demonstrated in other species [22,23,24].

A new recrystallized form of enrofloxacin, defined as hydrochloride-dihydrate (enro-C), has been characterized [25,26]. Enro-C shows considerably higher water solubility than the original compound, and it shows higher bioavailability than that of the original drug in dogs [27]. The C_MAX_ value after oral administration of enro-C was approximately three times higher compared to the enrofloxacin reference preparation. Consequently, a higher C_MAX_/MIC ratio can be obtained. Additionally, a higher C_MAX_/MIC ratio can be ensured by applying a gel containing 0.5% of enro-C to the affected skin, thrice per day. Considering the above, this trial aimed to assess the clinical efficacy of the combined therapy of orally administered enro-C, plus its topical application as a 0.5% enro-C gel in a defined group of dogs affected by deep-bacterial canine pyoderma, unresponsive to treatment (UDCP).

## 2. Material and Methods

### 2.1. Animals

All study procedures and animal care activities were carried out following the Institutional Committee for Research, Care, and Use of Experimental Animals of the National Autonomous University of Mexico (UNAM), under Official Mexican Regulation NOM-062-ZOO-1999 [28]. In this study, only dogs owned by the client (*n* = 55) were eligible. Dogs were recruited from three veterinary hospitals in Mexico City and those referred to the Pharmacology Department of the School of Veterinary Medicine from the National Autonomous University of Mexico (UNAM), also in Mexico City. Written informed consent forms were obtained from all owners. Animals under 12 (small and medium breeds) or 18 (large breeds) months old and pregnant or lactating females were also excluded.

### 2.2. Drug Preparation and Administration

Enro-C batches were prepared as indicated in Patent 472,715 (Mexico/Instituto Mexicano de Protección Industrial: IMPI MX/a/2013/014605 and PCT/Mx/2014/00192, Mexico City, Mexico). This process produces enrofloxacin hydrochloride-dihydrate with a purity of 99.97%. The original molecule of enrofloxacin chemical grade, was purchased from Globe Chemicals (Mexico). All dogs were weighed, and custom gelatin capsules were prepared accordingly and administered at a dose of 10 mg/kg/day orally, for as long as necessary according to the remission of signs (see Table 1). Treatment was ended when no signs could be detected as per Table 1 and Table 2. Additionally, a 0.5% enro-C gel was manufactured with 2% calcium alginate and 0.5% propylene glycol, and it was smeared onto dogs in affected areas thrice per day. Before the beginning of this trial, a 5-day washout period of previous medications was attempted in all cases but was not completed in 13 cases. No other anti-infective drug, antiparasitic, steroid therapy, non-steroidal anti-inflammatory drugs, or feed changes were allowed. During the test, weekly baths with a neutral-pH shampoo (Mennen zero%^®^, Colgate-Palmolive Company, Mexico City, Mexico) free from alcohol, dyes, silicones, perfume, selenium, fatty acids, and a pH = 6 were allowed.

### 2.3. Experimental Design

This study was conceived as an open-label longitudinal clinical trial. The dogs were divided according to their clinical signs in two groups with two severity grades of UDCP: severe and very severe. This was done according to the criteria presented in Table 2, which was based on formal literature [2,18,29,30]. The degree of severity of each sign was rated zero, 1, 2, or 3. Upon arrival at the clinic, a complete clinical history was obtained from owners, and the dogs were clinically examined and classified. Basal blood samples of 2–3 mL were obtained from all cases and were sent to carry out the corresponding renal, hepatic, and hematological profiles. These tests were repeated at the end of the treatment. Exclusion criteria were based on clinical signs, laboratory tests that indicated kidney or liver problems, and/or when the skin biopsy indicated other pathologies not considered as recurrent deep canine pyoderma, such as callus pyoderma, parasitic dermatoses, fungal infections, leishmaniasis, hyperadrenocorticism, growth hormone deficiency, diabetes mellitus, allergic pruritus, cutaneous neoplasia, juvenile cellulitis, calciphylaxis due to end-stage renal disease and hyperparathyroidism, immunomodulatory-responsive lymphocytic-plasmacytic pododermatitis, pemphigus foliaceous, pyoderma gangrenosum, and other unidentified pathologies [31]. All cases were followed every day until resolution. The classification of a case as treatment success was recorded when the criteria listed in Table 2 were met on time. Therefore, the decision of improvement or lack thereof was based solely on clinical signs. A two-month visual/clinical follow up was established after dogs were declared clinically cured, and no bacteriological follow up was attempted.

Because this study was an outpatient clinical trial, no dogs were hospitalized, so owners were instructed to disinfect and clean the dog’s habitat. They were also given an informational flyer to instruct them about the potential risks of inter-species infections/drug resistance, and how to minimize them. In addition to veterinary supervision, owners were asked to monitor any unwanted reactions in their dogs after each treatment, including allergic reactions, such as rash, increased pruritus, ataxia, depression, seizures, mood swings, changes in appetite, and any other manifestation of adverse reaction to the medication. This information should have been recorded and classified as an adverse event and treatment failure. The mean remission of the disease in dogs (days) was statistically analyzed using the Log rank-Mantel–Cox test.

### 2.4. Antimicrobial Susceptibility

Skin scrapes were performed to rule out scabies and demodicosis. Five bacterial swabs were obtained from the drainage of pustules, bullae, or fistulas. Disinfection of the outer part followed by deep swabbing (as deep as possible) was applied to reduce the risk of detecting only contaminating organisms. The samples were sent immediately to the Microbiology Department of the National Autonomous University of Mexico (UNAM) for bacteriological examination. Once there, the samples were transferred to a sterile tube with a screw cap containing 3 mL of modified Stuart transport medium. The culture was carried out in several media (e.g., blood agar, MacConkey agar, MSA, Rambach agar, and SS-agar) and then, it was incubated for 18–24 h at 37 °C. The identification of the species was performed by conventional methods [32], and the technique for identifying *Staphylococcus* sp. was that described by Kloos and Schleifer [33], and modified by Bannerman [34].

The recovered isolates were subjected to in vitro antibiotic sensitivity tests based on the broth-dilution method in microtiter plates containing 2-fold dilution amounts of antimicrobial compounds. Distinct panels were used for different bacterial species. The MIC breakpoints used in this study were adopted from published sources [35,36]. The ranges of antimicrobial concentrations in the panels were between 64 and 0.06 µg/mL. Resistant isolates were counted as such, as well as the intermediates readings. *Escherichia coli* ATCC 25,922 and *Staphylococcus aureus* ATCC 29,213 were used as quality control reference strains. There were no deviations from their expected results. Additionally, dermatophyte cultures were performed to exclude dermatophytosis.

## 3. Results

A total of 55 cases were admitted in this study, and they all met the inclusion criteria. Thirty-two dogs were classified as affected by severe UDCP and 23 dogs with very severe UDCP. Based on the set time-table to assume treatment success or failure (Table 2), it was observed that the combined effects of enro-C administrated orally plus topically, proved to be able to resolve all cases here included. Dogs with severe unresponsive pyoderma presented mainly papules, pustules, crusts, erythema, comedo, and pruritus, while dogs classified as affected by a very severe unresponsive pyoderma had the same signs plus cellulitis, nodules, fistulae, and slight fever (see Figure 1, Figure 2 and Figure 3). In 13 cases, the washout period was not completed due to the dog’s well-being, and the use of thyroxine was allowed in two dogs with confirmed hypothyroidism, although they were euthyroideal upon admission, due to their medication. Table 3 shows the clinical results of all dogs and a list of presumed causes for cases of severe and very severe unresponsive deep canine pyoderma. Adverse effects due to treatment with enro-C can be considered inconsequential and occurred in five dogs with mild hyporexia, seven cases in which loose feces occurred, and mild muscle pain and stiffness in a dog during the fourth week of treatment. However, all adverse events disappeared in two to five days after the withdrawal of the treatment. All signs became apparent to the owners and the vets during the third week of treatment but gradually disappeared after the end of their treatment. Hematological tests, blood kidney, and liver variables showed initial leukocytosis, an increase in absolute neutrophil and eosinophil counts, and high serum cholesterol levels. They also showed a slight decrease in hemoglobin concentration, and occasionally a slight decreased in the total erythrocyte count. At the end of the treatment, deviations from the reference values could not be detected. Figure 1, Figure 2 and Figure 3 show some examples of dogs treated with enro-C capsules and enro-C gel.

Statistical analysis of remission according to pyoderma severity was performed using the Log rank-Mantel–Cox test. Considering that this test is frequently used within clinical trials to establish the efficacy of a new treatment when the relevant variable is the time elapsed. In this case we confronted severe vs. very severe cases from the start of treatment until the subject had complete remission of the symptoms. These data are shown in Table 4 and Figure 4 as a cumulative proportion of dogs in total remission, according to the degree of initial severity. The average remission of the disease in dogs with severe lesions was 8 days, while for dogs with very severe injuries this was 12 days.

Forty-five percent of dogs with skin lesions classified as severe had a complete remission of their signs within the first 8 days of treatment, and on day 11, all dogs were completely cured. The 60% of dogs with skin lesions classified as very severe had a complete remission on day 12, and all of them were completely cured on day 16.

Microbiological findings are summarized in Table 5. Examination of the bacteriological culture of all 55 cases (five swabs from each case) resulted in the recovery of 99 bacterial isolates that could be considered as etiology. In nine dogs, no pathogens were identified. The main isolated pathogens were *Staphylococcus intermedius* (19/99), *Staphylococcus pseudintermedius* (16/99), *Staphylococcus epidermidis* (15/99), and *Staphylococcus pyogenes* (14/99); also, six cases of *Pseudomonas aeruginosa* were isolated. In almost all cases, mixed infections were not ruled out as etiology. Overall sensitivity was higher for enrofloxacin and amoxicillin-clavulanate and lower for methicillin, oxacillin, ampicillin, and cloxacillin. The isolates showed intermediate sensitivity to cefovecin and doxycycline.

## 4. Discussion

In this clinical trial no challenge was attempted, nor was a control group without treatment or another group treated with a different antibacterial drug preparation set. This decision was made upon the ethical considerations argued and sustained in previous clinical studies [37,38], and based on the owners’ refusal to be part of another treatment different from enro-C. All cases were studied in dogs from particular owners, and the study was designed as an open-label trial. Hence no information was withheld from the participating pet owners. Individual orientation was attempted, emphasizing the lack of flexibility in the dosing of enro-C, the need to avoid any concomitant medications, the timetable to expect results, their participation in detecting possible side effects, the availability of different treatments with other antibacterial drugs, and their availability for daily follow up, among other issues. The previous anecdotal clinical experiences with enro-C, shared by other owners, were the main drive to attend this study. Hence, owners rejected the option of including their pets in an experimental group, in which results would not meet their expectations. Additionally, this study was designed as longitudinal, based on the evaluation of cases with repeated observations of the same variables (skin lesions), and, in established periods. In this context, Paulus et al. [37] suggested that under some conditions, single-group studies provide useful information on the comparative effectiveness of interventions because there is an implicit comparison. For example, the expected course of the disease is known with almost certainty and the effect observed in the study group is evident, or the magnitude of the changes observed after treatment are indisputable. These circumstances apply to this study in which all dogs were carefully monitored to obtain a reasonable source of evidence. Within this context, it is worth emphasizing that the Mantel–Cox test can indicate the average and median times in which cases classified as very severe change to severe. This can be considered as an indication of clinical progress and emphasizes the adequacy of the classification used in this study. That time was very similar to the number of days in which severe cases needed to reach total remission: 8 days (test power = 0.857). Very severe cases of UDCP needed approximately an additional four days to show complete remission of the signs. The difference with severe cases of UDCP was statistically significant (*p* = 0.0001).

One year was needed to gather enough dogs for this trial. This can be partly be explained by the many available treatments and their considerable efficacy, particularly if fluoroquinolones are utilized [1,39]. Additionally, it is worth emphasizing that for this study, a careful selection of dogs was implemented. For example, demodicosis and dermatophytosis were ruled out since these diseases often cause hair loss, skin scrapes, and lesions that eventually induce pyoderma. It is unlikely that other skin diseases were present in the dogs studied here, such as atypical bacterial infections (e.g., actinomycosis or nocardiosis), autoimmune diseases, neoplasms, and so forth.

Despite the great efficacy obtained to treat deep canine pyoderma with various beta-lactams and fluoroquinolone derivatives, it has been reported that bacterial resistance is increasing [4] and the continued use of antibacterial drugs in the treatment of small animals is one of the leading causes of this phenomenon [1]. In this study, the sensitivity rate to commonly prescribed antibacterial drugs for isolated bacteria would appear lower than the one reported in studies from 12 years ago [40]. However MIC analyses of local pathogens should be carried out to clarify this observation, and preferably utilizing standards for veterinary medicine. Notwithstanding the above, results coincide with a more recent evaluation [41]. The combination of amoxicillin-potassium clavulanate and enrofloxacin proved to have the highest sensitivities, as previously described by Pedersen et al. [40] and Shah et al. [41], but it is contrary to other data [39]. The discrepancies can be explained by geographical trends in the use of some antibacterial agents as well as the particular laboratory techniques and standards utilized. For example, in a study conducted in Portugal to define the dermatological use of antibacterial drugs in small species, most veterinarians (57%) declared an increase in the number of antibiotic-resistant cases observed in the last five years for *Staphylococcus intermedius* [42]. Hence, it can be safely said that the treatment of pyoderma in dogs is increasingly linked to bacterial resistance worldwide. From Table 5 it is possible to state that the percentages of sensitivity to methicillin, oxacillin, and cefovecin, which are markers of methicillin-resistant *Staphylococcus aureus*/methicillin-resistant *Staphylococcus pseudintermedius* (MRSA/MRSP), are lower than the percentages of sensitivity to amoxicillin-clavulanic acid. Hence, based on these percentages, most of the isolated strains could be MRSA/MRSP pathogens. This observation deserves further definition in a separate work. Nevertheless, this was the clinical scenario chosen for this trial, and under such conditions, the combined effects of enro-C administered orally plus topically, proved to be able to resolve all cases of UDCP treated. In addition, it is important to relate positively to this outcome the susceptibility patterns found for enrofloxacin (from 31%–43%—Table 5), and the improvement observed in all dogs. It is true that to ensure the success of antimicrobial therapy, veterinarians often tend to use newer and/or broad-spectrum medications, such as fluoroquinolones or cephalosporins. Therefore, it would be commendable to prescribe the enro-C dual treatment here described, only after a positive antibiogram indicates, and its use should be limited to those situations in which other antimicrobial agents have not been able to cure.

The suitability of previous unsuccessful treatments administered to dogs included in this study, as well as the adequate compliance with the veterinarian’s instructions, the quality of the pharmacological preparations, and other similar considerations are beyond the objectives of this trial. However, a larger study is currently underway to try to obtain additional data on these issues.

As far as fluoroquinolones are concerned, they have been recommended for the treatment of canine bacterial pyoderma and a high success rate has been reported. In particular for enrofloxacin 93.3% efficacy was reported, but with a 25% recurrences [15]. Additionally, excellent results were obtained with 5 mg/kg/day of oral enrofloxacin in 85.2% of the dogs, fair results in 11.1%, and 3.7% showed no response [43]. More recently, fluoroquinolones such as ibafloxacin, which are not available in many countries, i.e., Mexico and South America, were administered orally daily at a dose of 15 mg/kg and required an average of 41 days to exhibit an efficacy of 74%, while marbofloxacin at a dose of 2 mg/kg/day for approximately 38 days had an 81% efficacy in treating cases of both superficial and deep pyoderma [14]. Pradofloxacin, a fluoroquinolone not approved for use in dogs because it induces bone marrow suppression (https://www.bayerdvm.com/products/veraflox-pradofloxacin-oral-suspension-for-cats/), was assessed in dogs (3 mg/kg/day PO) and it achieved an 86% efficacy after approximately 5 weeks of treatment with no recurrences after a clinical follow up of 11 weeks [13]. Considering this information, enro-C administered orally and topically, can now be added to this list of fluoroquinolones with high efficacy. However, confirmation to determine if there is greater efficacy of the dual treatment here proposed with enro-C over other treatments is still required. For example, it will be necessary to compare MIC values and/or mutant preventive concentrations (MPC) achieved in the affected skin areas as this feature guarantees superior clinical efficacy and lessens the emergence of bacterial resistance [44], a problem linked to low doses of all fluoroquinolones [45].

This study was not designed to find bacterial patterns of antibiotic resistance, but a complete bacteriological work should follow, as it can help explain why enro-C was able to achieve the great successes observed in the UDCP cases. From a pharmacological point of view, the exceptionally high values of C_MAX_ and AUC_0–24_ obtained by enro-C in dogs [27] might be part of the explanation. Additionally, the concurrent oral administration of enro-C plus its topical application as a gel in the affected areas can achieve particularly high concentrations of enrofloxacin in the affected skin, a proposal that also needs validation through experimental work. In general, fluoroquinolones should not be prescribed at low doses, particularly in dogs with allergic or endocrine skin diseases or when affected by UDCP, and that are frequently treated with these antibacterial drugs [45,46,47]. Based on this, a relatively high-dose of enro-C was chosen, supported by the topical smearing of the drug. The dogs selected for this trial had a history of unresponsive deep pyoderma, so a detailed study of the characterization of multi-drug resistant pathogens, such as methicillin-resistant *Staphyloccus* spp, could have better defined the role of enro-C in the treatment of such cases [1]. This work is currently being carried out. Nevertheless, efficacy assessment was considered as the first logical step. Up until just a few years ago, resistance to multiple drugs in causative pathogens had historically been rare in dogs. However, the incidence of resistance to antibacterial drugs is increasing, and *Staphylococcus epidermidis* and *Staphylococcus schleiferi* have been identified as causes of UDCP [48]. Hence, a dual treatment with enro-C or other fluoroquinolones may be useful to lessen the emergence of bacterial resistance.

## 5. Conclusions

Given the outstanding clinical responses observed in this trial it can be concluded that for enrofloxacin sensitive bacteria as shown in an antibiogram, the dual treatment with enro-C administered orally and topically shows a high efficiency in the resolution of cases of deep canine pyoderma not responsive to previous antibiotic therapy.

## Figures and Tables

**Figure 1 animals-10-00943-f001:**
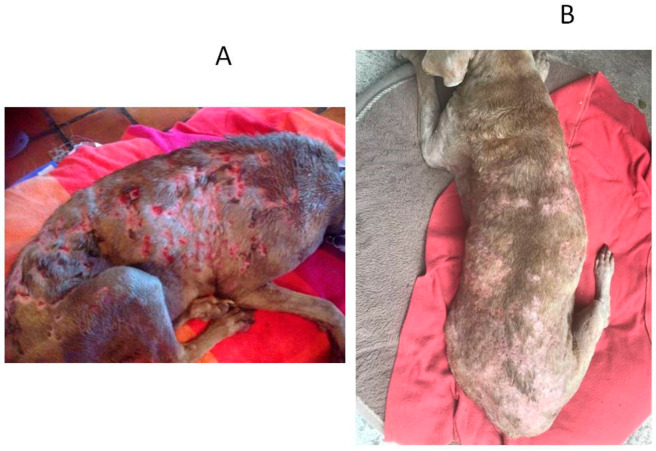
Canine unresponsive deep-pyoderma, classified as very severe, and treated with enrofloxacin HCl-2H_2_O (enro-C) at a dose of 10 mg/kg orally in gelatin capsules, plus the topical administration thrice a day of enro-C prepared as gel. Aspect of the disease in a dog at admission (**A**) and 10 days later (**B**).

**Figure 2 animals-10-00943-f002:**
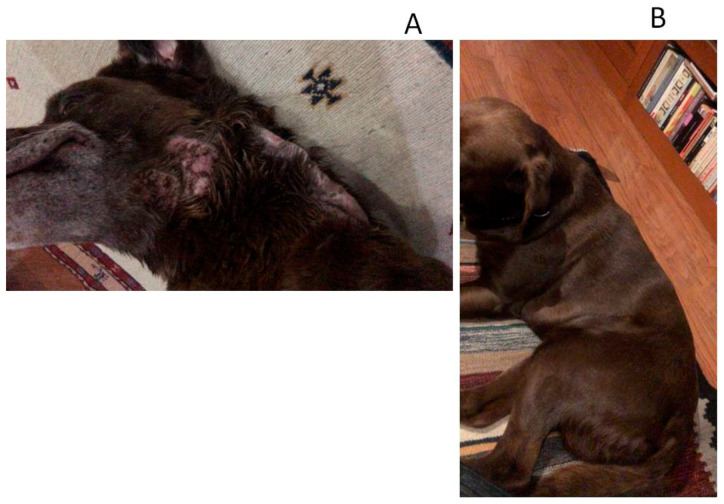
Canine unresponsive deep-pyoderma, classified as severe, and treated with enrofloxacin HCl-2H_2_O (enro-C) at a dose of 10 mg/kg orally in gelatin capsules, plus the topical administration thrice a day of enro-C prepared as gel. Aspects of the disease in a dog at admission (**A**) and after treatment (**B**).

**Figure 3 animals-10-00943-f003:**
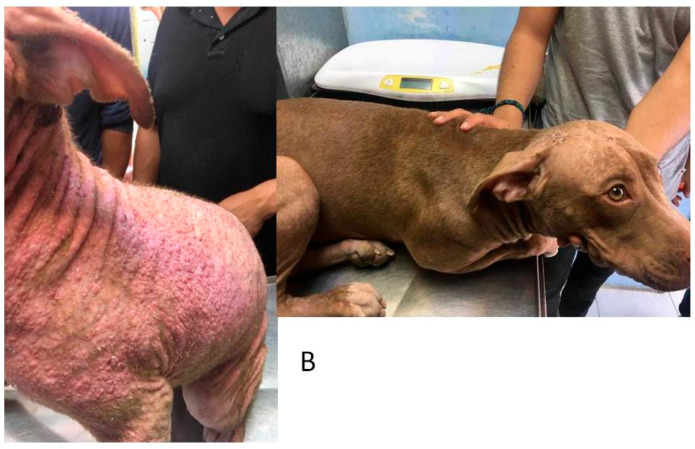
Canine unresponsive deep-pyoderma, classified as very severe, and treated with enrofloxacin HCl-2H_2_O (enro-C) at a dose of 10 mg/kg orally in gelatin capsules plus enro-C as gel, thrice a day. Aspects of the disease in a dog at admission (**A**) and 17 days later (**B**).

**Figure 4 animals-10-00943-f004:**
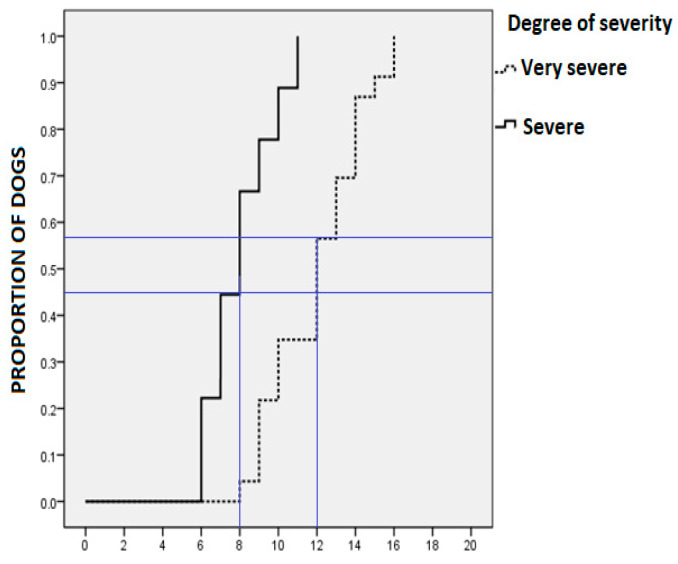
Cumulative proportion of dogs in total remission, according to the degree of initial severity of unresponsive deep-pyoderma.

**Table 1 animals-10-00943-t001:** Signs of positive response to treatment, and days in which they were expected to occur, in dogs with unresponsive canine deep pyoderma classified as severe to very severe. Dogs were treated orally with enro-C (10 mg/kg/day) plus an enro-C gel 0.5% applied topically on affected areas thrice a day.

Sign	Classification of Pyoderma	
	Severe	Very Severe
Pruritus cessation or marked reduction	≤3 days	≤5 (days)
Initial resolution of skin lesions	≤7 days	≤10 days
Initial fur growth	≤15 days	≤15 days
Absence of peculiar odor	≤3 days	≤5 days
Absence of other signs (fever, hyporexia, postural discomfort, etc.)	≤3 days	≤5 days
Absence of recurrence	2 months	2 months

**Table 2 animals-10-00943-t002:** Scoring system to classify unresponsive canine deep-pyoderma as severe or very severe, based on clinical signs (as zero, 1, 2, or 3).

Classification	Signs	
	Severe	Very Severe
**Pruritus**	Moderate to severe (2)	Severe-constant (3)
**Fever**	No (zero)	Slightly increased (3)
**Appetite**	Not affected—slightly reduced (0–2)	Greatly reduced (3)
**Papules**	Few—some ^#^ (1,2)	Many (3)
**Pustules**	Few—some ^#^ (1,2)	Many (3)
**Erythema**	Localized in few areas (2)	In almost the whole body surface (3)
**Crusts**	Few—some ^#^ (1,2)	Many (3)
**Comedo**	Few—Some ^#^ (1,2)	Many (3)
**Fistulae**	Few—Some ^#^ (1,2)	Many (3)
**Cellulitis**	Slight–moderate (1,2)	Marked (3)
**Estimated affected body surface**	≥50% (2)	≥75% (3)
**Total score**	From 10 to19 *	20–33

* Cases scoring less than 10 points were not included in this trial. ^#^ Papules, pustules, crusts, comedo, fistulae were graded as follows (1): very few lesions and they were hardly visible from 2 m away; (2): red lesions and inflammation that appear worthy of treatment keeping the same distance; and (3): loaded with these lesions and easily recognized at 2 m (adapted from Adityan et al. [29].

**Table 3 animals-10-00943-t003:** Summary of treatment outcomes of 55 cases of deep-canine pyoderma, unresponsive to initial antibacterial treatment, and treated orally with enrofloxacin HCl-2H_2_O (enro-C) at a dose of 10 mg/kg once a day and enro-C as gel, topically applied on affected areas, thrice a day.

Feature	Deep-Pyoderma	
	Severe	Very Severe
No of cases treated	32	23
No of days on treatment *	8.03 ± 2.1	12.0 ± 2.4
Time to control of pruritus	2.2 ± 0.6	3.8 ± 0.8
Adverse drug reactions º	none	Hyporexia cases 5 cases; loose feces 7 cases; muscle pain 1 case
Treatment success	100%	100%
possibly facilitated by:		
Dog-flea collar reaction	4	8
Flea allergy	3	6
Post-grooming furunculosis	8	5
After a dog bite or fight	2	1
Lesions in pressure points	2	2
Previous history of demodicosis	2	none
Unidentified	11	1
Previously treated with ^#, ∞^:		
Amoxicillin/K-clavulanate: (5:1) 20 mg/kg PO bid	12	8
Cefovecin: 8 mg/kg SC once every 7 days	9	8
Marbofloxacin: mg/kg PO, every 24 h	6	6
Enrofloxacin: 10 mg/kg PO, every 24 h	5	6
Cephalexin: 20 mg/kg PO tid	4	4
Clindamycin: 20 mg/kg PO bid	0	2

º Observed after the third week of treatment. * After clinical cure, treatment was extended for 5 days after identifiable remission of skin lesions. ^#^ Some dogs were treated with more than one antibacterial drug before entering this trial. **^∞^** Doses were recorded based on the manufacturers’ instructions and confirmed with the dog’s owner.

**Table 4 animals-10-00943-t004:** Log rank-Mantel–Cox statistical analysis comparing the remission time of dogs with severe and very severe skin lesions. This test allows the creation of a third set of results, i.e., the passage from very severe to severe, as an estimate of clinical progress.

Severity	Mean				Median			
	Value	Typical Error	95% Confidence Interval		Value	Typical Error	95% Confidence Interval	
			Lower Limit	Upper Limit			Lower Limit	Upper Limit
**Severe**	8.000 ^a^	0.577	6.868	9.132	8.000	0.707	6.614	9.386
**Very severe**	12.000 ^b^	0.495	11.030	12.970	12.000	0.951	10.136	13.864
**From severe to very severe**	8.043 ^a^	0.472	7.117	8.970	7.000	0.799	5.435	8.565
**Global**	9.691	0.399	8.910	10.472	9.000	0.494	8.031	9.969

^a,b^ Different letter indicate a statistically significant difference. X^2^ Log-rank/Mantel-Cox with 2 degrees of freedom = 27.62; *p* = 0.0001, 1 − β = 0.857, where β = probability of type 2 error, and 1 − β is the power of the test.

**Table 5 animals-10-00943-t005:** In vitro antibiotic sensitivity test of 46 isolates recovered from dogs affected by unresponsive deep canine pyoderma.

Isolate	No. of Isolates	Percentage of Isolates Sensitive to the Antibiotic Used										
		Am	AmC	Mb	Me	Ox	Amp	Cl	Cf	TmS	Dox	En
*Staphylococcus intermedius*	19	15.8	26.3	15.8	10.5	5.3	5.3	10.5	21.1	15.8	15.8	42.1
*Staphylococcus pyogenes*	14	14.3	42.9	7.1	7.1	7.1	7.1	0	7.1	14.3	21.4	42.9
*Staphylococcus aureus*	6	16.7	33.3	50.0	16.7	0	16.7	0	16.7	16.7	16.7	33.3
*Staphylococcus pseudintermedius*	16	12.5	31.3	12.5	6.3	12.5	12.5	6.3	12.5	12.5	18.8	31.3
*Staphylococcus epidermidis*	15	13.3	40.0	6.7	6.7	6.7	13.3	6.7	13.3	6.7	20.0	40.0
*Staphylococcus saprophyticus*	8	12.5	25.0	12.5	12.5	0	12.5	12.5	12.5	12.5	25.0	37.5
*Pseudomonas aeruginosa*	6	16.7	16.7	16.7	16.7	0.0	16.7	16.7	0.0	16.7	16.7	16.7
*Klebsiella* sp.	4	0	0	50.0	0	0	0	0	0	25.0	25.0	25.0
*Streptococcus* sp.	7	0	0	28.6	0	0	0	0	0	14.3	14.3	0.0
*Escherichia coli*	4	0	0	0.0	0	0	0	0	0	0	25.0	25.0

Am = Amoxicillin; AmC = amoxicillin-clavulanic acid; Mb = Marbofloxacin; Me = methicillin; Ox = Oxacillin; Amp = ampicillin; TmS = trimethoprim–sulfamethoxazole; CF = Cefovecin; En = enrofloxacin; Dox = Doxycycline; Cl = clindamycin.

## Data Availability

All datasets generated for this study are available upon request.

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
