# Peer review of "Oral Plus Topical Administration of Enrofloxacin-Hydrochloride-Dihydrate for the Treatment of Unresponsive Canine Pyoderma. A Clinical Trial"

_animals, 2020, doi:10.3390/ani10060943_

Round 1

Reviewer 1 Report

The article entitled "ORAL PLUS TOPICAL ADMINISTRATION OF ENROFLOXACIN-HYDROCHLORIDE-DIHYDRATE FOR THE TREATMENT OF UNRESPONSIVE CANINE PYODERMA. A CLINICAL TRIAL" is very interesting; however, there are many areas that need improvement. 

Introduction: It is not specific and provides very superficial information. The English need improvement to increase clarity. Pyoderma is a secondary infection associated to an underlying primary cause, not the other way around (line 52). I do not see the point to list the underlying causes; however, if the authors want to do that, they should focus on the most common causes and not the rarities. Similarly, other causes of lack of an antibiotic response are not listed (line 60) (e.g. wrong dose, wrong antibiotic, lack of identify underlying cause, owner compliance). In addition, fluoroquinolones should be used based on culture and sensitivity and reserved to Gram-negative bacteria if possible; other antibiotics are available for Staphylococci. This aspect is not mentioned at all. The authors state a hypothesis that does not match with the study. They are assuming that they reach a higher cutaneous concentration of enrofloxacin, but they did not prove that. Finally, change RDCP in UDCP (line 92 and 114).

M&M: This section needs some improvements in term of details. As reported, the study is not repeatable. Important clinical information on the dogs is missing (e.g. what type of lesions were sampled, what antibiotics were used unsuccessfully, what were the underlying causes of the pyoderma, what was the rate of reoccurrence of the pyoderma, age, sex, etc.). Those are essential pieces of information to better identify and characterize the pyoderma and the degree of “self-resolution”. What do the authors mean with “exclusion criteria were based on clinical signs” (line 113)? What are these clinical signs? The paragraphs on lines 140-147 and 154-158 should be moved before the microbiology section (line 120). The sample methodology is not clear. What do the author jean for “deep swabbing”? Specifically in the case of papules, or nodules?

How were the bacteria identified (indicate the microbiology system used)? The references for the MIC are very old (2002-2004), I would encourage the authors to update the results in the face of the most updated CSLI guidelines. In addition, the clinical microbiological laboratory is a human or veterinary laboratory. There is a significant difference in microbiological identification of veterinary microorganisms when reported by a human laboratory.

What shampoos were the owners using (line 156)? Were they applying anything else? Please spell out AINE (I am assuming the authors meant NSAID).

As far as enro-C preparation, more information is needed on the enrofloxacin compound. Was the purchase enrofloxacin medical or chemical grade? The dose of 10mg/kg/day was administered every 24 hours orally or divided? Are the 10mg/kg/day referring to the mgs of basic enrofloxacin?

Results: The clinical description of the lesions should be checked (“scabs” is not a medical terminology and crusted papules are described in both severe and very severe, so are not different lesions). How long did it take the adverse effects to disappear, how pronounce they were (lines 185-189)? As far as the hematological changes, in how many dogs did they occur? Please change “injuries” to “lesions” in line 226.

I am not sure what the point of the Mantel-Cox analysis is. Being this an uncontrolled study, there is no comparison between groups. Unless the authors are trying to compare the time of resolution between severe and very severe cases, in which case, this needs to be better explained.

As far as the microbiological results, the text is not in agreement with the table 5. More specifically, they report 99 isolates (out of 55 dogs), but in the table, they refer to 46 isolates; please clarify. In addition, it is important to relate these results with the previous history since >50% of isolates were susceptible to enrofloxacin. For this reason, it does not surprise that the dogs improved.

Finally, no information is provided on eth follow up period of 2 months. How many dogs relapse or what happened after 2 months of therapy?

Discussion: The discussion is very long. In the first section, the authors explain their reasoning more than compare and contrast with previous studies. They justify the uncontrolled study because of the ethical reasons, why did they not sue shampoo therapy as control? There are plenty of studies using shampoo therapy versus oral therapy in the literature. That type of study would have been much more effective. The authors specify that owners had strong expectations on the product and this was requested as criteria for accepting to enter the study. This type of scenario is very risky since opens a big door for selection bias and should be justified. On line 283, the authors comment on the failure of previous treatments for this cohort of dogs, was enrofloxacin used before? If so, what dose and did it fail? On lines 288-291, the authors exclude other underlying cause for these dogs, so were these cases of primary pyoderma? What about allergies (not only flea) and endocrinopathies as underlying cause? I would also suggest rewording the statement on lines 312-315, those are actually essential piece of information to validate and better understand this study and as such should be reported here and not in a future report. On lines 330-334, “high concentration “is referring to enrofloxacin or enro-C? Because 10mg/kg/day could be arguably considered a low dose for enrofloxacin. In addition, are the authors sure that the statement on lines 340-342 reflects the reference 38?

Conclusions: The conclusion statement is very strong since it is very likely that these dogs responded because the bacteria were or susceptible to enrofloxacin so maybe the enro-c did not make any difference or because of the topical administration. In fact, it is well known that topical antibacterial works much better than systemic because of the availability to apply a much higher concentration of antibiotic otherwise toxic to the patient.

Author Response

Best Regards,

Hector Sumano 

Reviewer 2 Report

Manuscript ID: animals-767350

This manuscript describes the efficacy of a combination of oral and topic administration of enrofloxacin-hydrochloride-dihydrate for the treatment of unreposnsive canine pyoderma. After careful review of the present manuscript, in my opinion, in the current form the article cannot be accepted for publication, that is why I suggest some major revisions in order to improve the quality of the paper.

General comments:

The information provided is of some interest, but overall I found some lacks in the experimental design and consequently in the results and their interpretation. Additionally, this paper could somehow contribute to new insights into the knowledge of this specific topic, especially from a clinic point of view, but it needs to be re-organized. The grammar could be polished throughout the manuscript for ease of reading and I think the English language needs some little editing.

Abstract: it gives some general information but it has to be improved after a major revision of the paper itself.

Introduction

Pag. 2 Line 68: S. pseudintermedius. In general check carefully the spelling of all the bacteria through the whole manuscript (i.e Pseudomonas aeruginosa) and the microorganisms must be in italics.

Pag 3 Line 88-92. The aim of the work is not clear at all. Authors must correctly specify the objectives of their work at the end of the introduction.

Material & Methods

Pag. 3 Line 106-108The dogs were divide in two groups (severe and very severe) according to clinical evaluation. Is there a reference? It is a clinical evaluation, so somehow empiric, so you need to add a reference of a validated method.  For example in Table 1 is shown the scoring system: pustuels if “few-some” is classified as severe, if “many” is classified as very sevre. The sampe for crusts, comedon, papules and so on… This is your inclusion criteria, I mean in the rest of your paper you divided your population in severe and very severe, so you really need to be more specific here, adding references.

Line 114 Spell RDCP

Pag 4 Line 127, delete “to support the growt of E. coli, Staphylococcus spp …” The media you used in lab are common media used for all the bacteria so you do not really need to specify this sentence.

Line 128: How did you identify Staphylococci at species level?

Line 137: the intermediates isolates are usually counted among the resistant ones not the susceptible

Pag 5 Line 148. This paper aims to promote the oral plus topic combination of enrofloxacin-hydrochloride-dihydrate for the treatment of unreposnsive canine pyoderma (it is even the title). I would have expected a clear scheme in the methods about the antibiotic dosage, administration, time and so on. Instead, it appears first in the legend of Table 2 (10 mg/kg/day). This point needs to be clarified because it is crucial.

Line 152 Delete Pyometra in the table and replace it with Pyoderma

Line 156 What do you mean with AINE? Fans? Use the English spelling

Line 165-166 It is not clear the topic therapy: again the fact that it was twice a day is only written in the legend of table 2. The therapeutic scheme in the 55 dogs must be clarified.

Results

The results are not clear. First of all see the first comment on severe and very severe pyoderma. Then, Fig 1-2-3 do not mention at all the topic therapy.

Pag 10 Table 4 is not clear and it is now well explained in the text.

Pag 11 Table 5. Where 46 isolates came from? I would suggest to present these data as histogram. Again check the spelling of the bacterial name. Typo in Percentage in the table. Then, in the abstract you reported just some of the bacteria found but it is no clear why you chose you report just them. There are al least 3 o 4 bacteria before Pseudomonas (i.e Streptococcus strains are 7 for example).

Then, how do you say “S. intermedius” or “S. pseudintermedius” or “S. aureus”. These 3 bacteria are phenotypically identic in plate and even other characteristics such are coagulase test is not discriminatory. So you need to clarify (in the methods) what test you performed otherwise there results are not acceptable.

Discussion

The discussion section should be improved and totally reorganized, only after you improve the rest of the papere. Firsty: start with specify clearly the objective of your study. Then make sure the methods are correct to prove your experimental hypothesis. Male your results clear. At that point it will be more easy for you discuss your data.

Pag 12 Line 294-296 You can’t say that because you did not report the MIC results

Line 305 you said in the discussion “the combined effects of Enro-C administrated orally plus topically  proved to be able to resolve all cases of UDCP”. Well, this also is not clear reading the paper. You should make easier the reading, especially underlining the key point if your results in a very clear way.

Line 309-311 Please specify what you meant.

Additionally, I suggest to the authors, even to underline the lacking points of their work (th absence of a control group for instance) or the necessity to validate your results through further analyses.

Such a paragraph would not detract from the quality of the work, rather it would demonstrate this potential limitation and consequent caution in interpretation of your results.

Author Response

Best Regards,

Hector Sumano 

Reviewer 3 Report

The authors deal with an important subject which is the treatment of dog pyoderma. In the current context of antibiotic resistance, having products that are used for a shorter period of time and that are effective is potentially very useful information for the practitioner.
Thus, this new form of enrofloxacin seems very promising and the study that has been conducted involves a large number of animals (55). Nevertheless, several important flaws are present in this manuscript, making the study unpublishable.
1- the authors seem to lack knowledge of dog pyoderma, both in their pathogenesis and in their description. Indeed, they seem to consider most pyoderma as primary; however, the opposite is true: primary pyoderma is rare, even extremely rare. Moreover, they refer to a degree of severity but at no time do they cite the classification as used in veterinary dermatology: superficial pyoderma or folliculitis and deep pyoderma or furunculosis/cellulitis. Thus the introduction should be reviewed as well as the distribution of animals and the clinical inclusion criteria. What is the basis (validation?) of the scoring system used?
2- The authors seem to be unaware of, or in any case almost never refer to (except at the end of the discussion section), methicilin-resistant staphylococci, germs resistant to all Blactams . Table 5, which summarizes the results of the sensitivity tests, is very inconsistent. Indeed, the percentages of sensitivity to methicilin, oxacillin, cefovecine which are markers of MRSA/MRSP resistance are lower than the percentages of sensitivity to amoxicillin-clavulanic acid. Based on these percentages, most of the strains they have isolated are MRSA/MRSP ???
No indication is provided as to the method of identification of the bacteria. In addition, it is surprising to see both S. intermedius and S. pseudintermedius. The methods and results are therefore completely to be reviewed.
3- The methodology for monitoring the animals is lacking: no cytological examination is carried out (neither before nor during the study). The reader is unaware of the frequency and duration of follow-ups as well as the duration of treatment.
4- The excellent results reported are unusual in this type of disease. No indication is given on the concomitant treatments; in particular, the reader does not know if antiparasitic molecules (external or internal) were administered? if the food was up-graded or not? if adjuvant topical treatments (as is usual in this type of pyoderma) based on antiseptics were applied before, during or after the treatment. These elements (antiparasitic, food) could explain a better immune competence.
Figures 1 to 3 showing the appearance of the dogs before/after are very surprising, especially figure 3: the rapid and almost complete regrowth of hair in 7 days is very surprising...!
5- How long after the treatment was stopped were the animals monitored? In fact, the problem in this type of pyoderma is not so much the control of an episode as the control of recurrences.On what criteria is cure defined? visual aspect? palpation-pressure of lesion sites? negative bacteriology?

Author Response

Best Regards,

Hector Sumano 

Round 2

Reviewer 1 Report

The authors have replied to all the questions raised; however, the manuscript is still hard to follow and read. More specific details on the statistical software used are missing. In addition, no information is reported about the follow up as requested in the revision 1. The discussion has been completely changed, more pertinent and shortened. Although a great deal of explanations is provided regarding the reasoning behind the open design decided for this study; this section could be shortened.
Tables and figures
I am not sure about the usefulness and interpretation of the table2. Do the authors mean that those are expected results?
The figures could be improved. The dogs are in different positions and do not show the same area of the body before and after treatment. Figure 1: lesions on right side before, but side left for after treatment. Figure 2: Lesions on side left of chin and neck, but dorsum from after treatment. Figure 3 lesions on left side, but right side of the body after treatment.
Table 3: The dose of marbofloxacin is not present. Are the authors sure about the dose of clindamycin (20mg/kgq12hours)?
Looking at the table 5, I noticed that the authors report a sensitivity to doxy for Gram- bacteria ranging from 16.7 to 25%, are the authors sure about these data?
Finally, why the authors picked the 45% as end mark?

Reviewer 2 Report

The changes made to the manuscript significantly strengthen it. I think it is more readable and easier to follow. The authors addressed all major concerns. A few minor concerns are listed below:

English spelling through the whole manuscript need to be cheked. For instance chane all the "trice" in "thrice" in the summary, line 85, line 172, all the figure legends and so on

Again, check the spelling of bacteria name, it is not pseudointermedius but S. pseudintermedius.

In the abstract as weel as in the bacteriological results I still do not understand why you decided to report these germs. I mean, if you want to list just the main ones delate P. aeruginosa. If you want to list all of them you have to write S. saprophyticus(8/99), Streptococcus sp (7/99), and so on...

Line 85 Delete "considering above"

Line 87 (UDCP) has to be just after the deep-bacterial canine pyoderma

As it is quite important that other primary disease have been excluded in the diagnose of UDCP I strongly suggest to declare in MeM that the dogs were vistited by a specialised dermatologist vet, who exluded allergic or endocrine disease or other conditions that can mask the follow-up results.
